# FewBodyToolkit.jl: a Julia package for solving quantum few-body problems

**Lucas Happ**

Few-body Systems in Physics Laboratory, RIKEN Nishina Center, Wakō, Saitama 351-0198, Japan

lucas.happ@riken.jp

## Abstract

Few-body physics explores quantum systems of a small number of particles, bridging the gap between single-particle and many-body regimes. To provide an accessible tool for such studies, we present FewBodyToolkit.jl, a Julia package for quantum few-body simulations. The package supports general two- and three-body systems in various spatial dimensions with arbitrary pair-interactions, and allows to calculate bound and resonant states. The implementation is based on the well-established Gaussian expansion method and we illustrate the package's capabilities through benchmarks and research examples. The package comes with documentation and examples, making it useful for research, teaching, benchmarking, and method development.

# 1  Introduction

As the name suggests, few-body physics studies physical systems of a few particles, usu-
ally between two and ten. The classical (gravitational) few-body problem has a century-
old history [1–3]. The present work, however, focuses on quantum systems governed by the
Schrödinger equation. Interest in quantum few-body systems emerged soon after the formu-
lation of quantum mechanics: a natural extension of the hydrogen atom is the molecular ion
$H_2^+$, a genuine three-body system [4]. Few-body calculations also provide microscopic input
for many-body phenomena, for example three-body losses in gases of ultracold atoms [5]. In
addition, phenomena unique to few-body systems, such as the Efimov effect [6,7] and related
universality, have helped establish a distinct few-body research community.

There exist several software packages [8–18] to simulate quantum systems, in particu-
lar for single-particle problems or two-body systems, partially with impressive visualization
capabilities. Yet, they rarely target genuine few-body systems beyond two particles. To our
knowledge, no general few-body solver with both accessibility and comprehensive documen-
tation is currently available. With **FewBodyToolkit.jl** we aim to fill this gap by providing an
open-source Julia implementation for solving general few-body problems.

For this we make use of the Gaussian expansion method [19], a well-established method
used by many researchers in the few-body community across hadronic, nuclear, and atomic
physics [20–28]. For this package we adopt it to two-body and three-body problems in vari-
ous dimensions (1D-3D), for pair-interactions of arbitrary shape, bound states or resonances,
and calculation of observables (e.g. mean square radii). Moreover, the package provides au-
tomatic computing of angular momentum coupling and (anti-) symmetrization. The current
feature set is motivated by recent research on few-body systems, e.g. on low-dimensional
systems [29–36], or few-body resonances [27,28,37–40], among others. Therefore, the idea
for this package originates from the unification of numerical codes developed and used in the
author's previous research [23,27,28], now provided under a common API. Despite the at-
tempt for generality, there clearly remain many possibilities for extensions, such as arbitrary
particle number, types of interactions, external fields, or entirely different methods. While
being motivated by and focused on research, we see further applications for teaching, as an
entry point for researchers from other communities, as a basis for new method development,
or as a benchmark tool. The API is deliberately kept simple, with documentation and runnable
example scripts to lower the barrier for new users. The package is implemented in Julia, lever-
aging features such as multiple dispatch and parametric types to combine performance with
flexibility.

The present article is organized as follows: First, we introduce basic few-body notations
and the implemented method in Section 2. Then, we present the current features of our
package (Section 3) and its implementation details (Section 4). We follow it up by some
examples in Section 5 as well as benchmarks in Section 6, and finally conclude in Section 7.

## 2 Method

In this section we provide a brief introduction to the coordinate systems and equations commonly used to describe few-body systems. Moreover, an outline to the Gaussian expansion method, the method employed in this package, is presented.

### 2.1 Coordinates and Schrödinger equations

In studies of few-body systems it is common to assume that the center-of-mass motion separates from the internal dynamics. This assumption is justified as long as external forces are absent or harmonic. When this condition is not fulfilled, an approximate description using harmonic confinement is often still suitable. In the present work we adopt this assumption, which allows us to restrict the description to the internal dynamics in the center-of-mass frame.

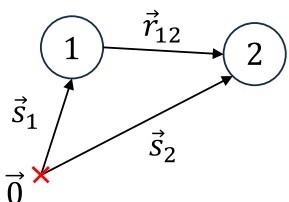

(a) Relative coordinate for two-body systems.

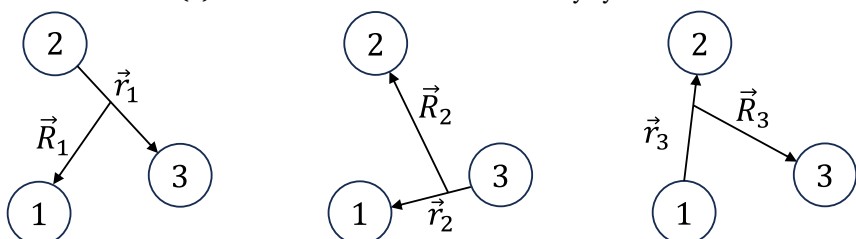

(b) Three sets of Jacobi coordinates for three-body systems.

Then, a two-body system can be described by a single relative coordinate $\vec{r}_{12} = \vec{s}_1 - \vec{s}_2$, as seen in Fig. 1a. To describe three-body systems in their center-of-mass frame, one can extend this scheme by using Jacobi-coordinates, see Fig. 1b. Here, the full system is described by two relative coordinates: $\vec{r}$, the direct relative coordinate between a pair of two particles, and $\vec{R}$, the vector from the pair's center-of-mass to the third particle. However, there is a subtlety: there are three equivalent sets of these Jacobi coordinates, related to the three different partitions of three particles into a pair of two, and a single one, as displayed in Fig. 1b. It is therefore common to use the notation

$$\vec{r}_i \equiv \vec{s}_j - \vec{s}_k, \qquad \vec{R}_i = \vec{s}_i - \frac{m_j \vec{s}_j + m_k \vec{s}_k}{m_j + m_k} \tag{1}$$

with $\{i, j, k\} = \{1, 2, 3\}$ and the cyclic permutations $\{2, 3, 1\}$, $\{3, 1, 2\}$.

The quantum two- and three-body systems are then governed by the Schrödinger equations

$$\left[ -\frac{\hbar^2}{2\mu_{12}} \nabla^2_{\vec{r}_{12}} + V(\vec{r}_{12}) \right] \phi(\vec{r}_{12}) = E_2 \phi(\vec{r}_{12}) \tag{2}$$

and

$$\left[ -\frac{\hbar^2}{2\mu_{ij}} \nabla^2_{\vec{r}_k} - \frac{\hbar^2}{2\mu_k} \nabla^2_{\vec{R}_k} + V_{12}(\vec{r}_{12}) + V_{23}(\vec{r}_{23}) + V_{31}(\vec{r}_{31}) \right] \Psi(\vec{r}_k, \vec{R}_k) = E_3 \Psi(\vec{r}_k, \vec{R}_k) \tag{3}$$

with the reduced masses

$$\mu_{ij} = \frac{m_i m_j}{m_i + m_j}, \qquad \mu_k = \frac{m_k(m_i + m_j)}{m_k + m_i + m_j}. \tag{4}$$

Due to the three possible ways of partitioning, it is therefore common in few-body physics to decompose any given three-body state into a sum

$$\Psi(\vec{r}, \vec{R}) = \Psi^{(1)}(\vec{r}_1, \vec{R}_1) + \Psi^{(2)}(\vec{r}_2, \vec{R}_2) + \Psi^{(3)}(\vec{r}_3, \vec{R}_3) \tag{5}$$

of Faddeev components $\Psi^{(i)}(\vec{r}_i, \vec{R}_i)$ (sometimes called rearrangement channels), each described in a different Jacobi set $\{\vec{r}_i, \vec{R}_i\}$. In case some particles do not interact, or two or more are identical, the number of Faddeev components can be reduced. The code does this automatically.

## 2.2 Gaussian expansion method

The FewBodyToolkit.jl package implements the Gaussian expansion method to solve Schrödinger equations in the form of Eqs. (2), (3). For two-body systems this means that an unknown state is expanded into a set

$$\phi(\vec{r}) = \sum_\alpha c_\alpha \phi_\alpha(\vec{r}) \tag{6}$$

of Gaussian basis functions $\phi_\alpha$. Depending on the dimensionality they are defined as

$$\phi_\alpha(\vec{r}) = N_{n,l}\, r^l e^{-\nu_n r^2} f_{l,m}(\hat{r}), \qquad f_{l,m}(\hat{r}) = \begin{cases} Y_{l,m}(\theta, \phi) & ,3D \\ e^{im\phi} & ,2D \\ 1 & ,1D \end{cases} \tag{7}$$

with an appropriate factor $N_{n,l}$ to ensure normalization $\int \mathrm{d}r^d |\phi_\alpha(\vec{r})|^2 = 1$. Here, $Y_{l,m}$ denote the spherical harmonics.

Since Eq. (6) is a coherent superposition of Gaussian basis states, it can also represent the much wider set of non-Gaussian states [41, 42]. In principle this form of the radial wave functions is sufficient, however representing oscillatory states, e.g. highly-excited states or resonances, requires a large number of basis functions. In this case an extension to Gaussians with complex-valued [43] ranges $\nu \to \nu(1 \pm i\omega_{\mathrm{cr}})$ can be employed. This effectively extends the basis functions by an additional factor of $\sin(\nu\omega_{\mathrm{cr}}r^2)$ or $\cos(\nu\omega_{\mathrm{cr}}r^2)$, which can greatly enhance coverage of oscillatory states. At the same time, most analytical expressions for the code remain unchanged, and only the Gaussian range $\nu$ needs to be treated as a complex number.

For three-body systems one proceeds in a similar fashion. However, since there are three sets of two relative Jacobi coordinates, each Faddeev component is expanded into a set

$$\Psi^{(i)}(\vec{r}_i, \vec{R}_i) = \sum_\alpha c_\alpha \psi_\alpha^{(i)}(\vec{r}_i, \vec{R}_i) \tag{8}$$

of basis functions $\psi_\alpha^{(i)}$ which itself are composed of products of two functions

$$\psi_\alpha^{(i)}(\vec{r}_i, \vec{R}_i) = \phi_\alpha(\vec{r}_i)\Phi_\alpha(\vec{R}_i). \tag{9}$$

These functions are each defined as in the two-body case

$$\phi_\alpha(\vec{r}) = N_{l,m} r^l e^{-\nu_n r^2} f_{l,m}(\hat{r}) \tag{10}$$

$$\Phi_\alpha(\vec{R}) = N_{L,M} R^L e^{-\lambda_N R^2} f_{L,M}(\hat{R}) \tag{11}$$

where now $\alpha = \{n, l, N, L\}$.

Employing this basis, the Schrödinger equations can be cast in the form of a generalized matrix eigenvalue problem

$$H\vec{c} = ES\vec{c}, \qquad H = T + V \tag{12}$$

for the coefficient vector $\vec{c}$, with $T$ and $V$ denoting the matrices of the kinetic and potential energy operators, and $S$ the norm-overlap matrix. Their matrix elements are computed via $\langle \psi_{\alpha'}^{(i)} | \mathcal{O} | \psi_\alpha^{(j)} \rangle$ for the various operators of the kinetic energy, interaction, and overlap (for the overlap we have simply $\mathcal{O} = 1$), and accordingly for the two-body systems. The Gaussian basis is non-orthogonal, leading to dense matrices $H$ and $S$ and therefore to a generalized eigenvalue problem instead of a standard one. However, this is often compensated by the fact that many expressions can be derived analytically, which minimizes the amount of heavy numerical integrations.

More details and derivations on the method can be found in the review article [19], however focused on 3D systems.

## 3 Features

### 3.1 General Features

**FewBodyToolkit.jl** is available as a registered Julia package that can be installed with a single command, making it straightforward to install and use. Moreover, to support accessibility and reproducibility, the package provides **documentation** together with example scripts.

The package features three solver modules, each dedicated to a different class of few-body quantum problems, distinguished by the number of particles and the dimensionality of the system. For some potentials, analytical formulae for the matrix elements can be used (currently: Gaussian, 1D contact interaction), otherwise the package resorts to numerical evaluations. Additional potentials with analytical matrix elements can be implemented as separate types and making use of Julia's multiple dispatch. In the following subsection we list the module-specific features in more detail.

### 3.2 Module-specific Features

| Feature | GEM2B | GEM3B1D | ISGL |
|---|---|---|---|
| Number of particles | 2 | 3 | 3 |
| Spatial dimension | 1D, 2D, 3D | 1D | 3D |
| Range of Gaussians | real, complex | real | real |
| Bound states | Yes | Yes | Yes |
| Resonances via CSM | Yes | Yes | Yes |
| Wave function output | Yes | No | No |
| Observables calculation | No | No | Yes |
| Potentials (Symmetry) | (Central) Symmetric | Any | Central symmetric |
| Potentials (Shape) | Arbitrary | Arbitrary | Arbitrary |

Table 1: Overview of shared features across the modules of FewBodyToolkit.jl.

The notation "arbitrary" shape of the potential means that any general function $f(r)$ can be used as input, as long as the boundedness of the interaction matrix elements is ensured. This also allows to treat potentials which were obtained from numerical calculations, as long as an interpolated function is provided as input. In most cases, the requirement is that $f(r \to 0) \propto r^{-\alpha}$,

and $\alpha < d + l + l'$, with the dimensionality $d$ and $l, l'$ being the angular momenta of the bra and ket basis functions corresponding to the respective relative distance in which the interaction acts. The large-distance integrability is usually guaranteed by the strong suppression of the Gaussian form of the basis functions. In addition to the common criteria, the modules have the following specific features:

**GEM2B: Two-body solver**   Since the two-body solver can handle (central) symmetric potentials, a fixed but arbitrary angular momentum channel can be chosen. Moreover, the two-body module offers functions for basis-state optimization and an option to solve the inverse problem (provide the energy, find the corresponding potential strength). Finally, a separate solver function exists for coupled-channel problems including derivative operator terms. This is particularly helpful in typical Born-Oppenheimer or hyperspherical treatments.

**GEM3B1D, ISGL: Three-body solvers (1D, 3D)**   The three-body solvers contain an automatic decomposition into the necessary number of Faddeev components (rearrangement channels), respecting possible reductions due to identical particles. Moreover, the coupling of angular momenta (in 1D: parity waves) between the 2+1 partitions is done automatically for each Faddeev component and respects the symmetry of the particles and global quantities (parity, total angular momentum), if conserved. Due to the simpler angular momentum algebra, the 1D solver can handle parity-violating potentials, whereas the 3D solver is currently restricted to central symmetric potentials.

## 4   Implementation

In this section, we discuss the implementation in more detail. As already mentioned in the previous section, the package provides an implementation in three separate modules: `GEM2B`, `GEM3B1D`, `ISGL`. This modular design allows users to select the appropriate solver for the particle number and dimensionality.

### 4.1   Inputs and Outputs

The core part of each of the modules is a solver function (`GEM2B_solve`, `GEM3B1D_solve`, `ISGL_solve`). These functions have two main arguments: `phys_params`, for the physical parameters of the system (masses, interactions, parity, etc.), and `num_params` for the numerical parameters (number and range of Gaussian basis functions, etc.). Moreover, depending on the module there are several optional arguments, e.g. enabling complex scaling [44] to compute resonances. The full list of arguments is explained in the API reference in the **documentation**. The output always contains the eigenvalues, and optionally eigenvectors and computed mean values of observables.

### 4.2   Workflow

The core solver routines in **FewBodyToolkit.jl** follow a modular sequence of steps described below:

**1. Input validation**   Initial checks ensure consistency of masses, symmetries, and potential parameters to catch errors early.

**2. Size estimation**   The total size of the basis and required memory allocation is obtained. For three-body systems, the allowed angular momentum channels of the subsystems are determined based on globally conserved quantities. Moreover, in case of identical particles, a reduction of the number of considered Faddeev components is performed.

**3. Preallocation**   To avoid repeated reallocations, memory is allocated once for arrays used in intermediate calculations (Hamiltonian matrix, normalization, arrays for interpolation, etc.) and for the outputs (energy array, eigenvector matrix, etc.).

**4. Precomputation**   In order to improve performance, repeated evaluations of commonly used expressions (Factorials, Clebsch-Gordan coefficients, transformation between Jacobi coordinates, etc.) are avoided by caching results.

**5. Interpolation**   For three-body calculations, the total number of matrix elements to calculate can easily reach the ten-thousands, or millions. If no analytical formula exists (as is the case for most interaction potentials), one has to resort to numerical calculations, which quickly becomes costly for such numbers. To avoid this, an interpolation technique is employed. Due to the use of Gaussian basis functions, most interaction matrix elements boil down to integrals of the form $\int \mathrm{d}r\, r^{l_{\mathrm{eff}}} e^{-\alpha r^2} V(r)$, which has a smooth dependence on the Gaussian parameter $\alpha$. Hence, numerical evaluation is performed only for a fixed number (defined by the parameter `kmax_interpol`) of $\alpha$ values and subsequently interpolated over the required range of $\alpha$ values. In contrast, for two-body problems a much lower number of basis functions and matrix elements must be considered. Hence, the interpolation procedure is employed only for the three-body modules.

**6. Matrix filling**   The matrices for the kinetic energy, interaction, and overlap are filled based on analytical formulae or interpolated numerical integrations. Moreover, the total matrices are assembled by respecting the relevant Faddeev components.

**7. Generalized eigenvalue problem**   Due to the non-orthogonality of the basis functions, the essential step boils down to solving a generalized eigenvalue problem. The eigenvalues, and optionally eigenvectors, are obtained by a special two-step solver. This solver catches possibly ill-posed problems caused by numerical parameters that yield high overlap between the individual basis functions.

**8. Observables calculation**   When supported by the module (currently only ISGL), physical observables such as mean-square radii for the $r$ or $R$ variable, as well as any central observable $\mathcal{O}(r)$ depending solely on $r$ are computed from the resulting eigenvectors.

## 5 Examples

### 5.1 Installation and first run

Since this is a Julia package, we assume Julia to be installed. First-time users can install Julia via `curl -fsSL https://install.julialang.org | sh` from the command line. For more information, see https://julialang.org/install/. Then, Julia can be started via `julia` from the command line.

This package is listed and registered as an official Julia package. Therefore it can be installed and loaded via Julia's package manager. Before usage, the package must be loaded once in each new Julia instance (e.g. after restarting Julia).

```
using Pkg
Pkg.add("FewBodyToolkit") # installing the package

using FewBodyToolkit # loading the package
```

For a first run, try the example from the readme page:

```
using FewBodyToolkit

# Define physical parameters: interactions and masses
v12(r) = -10/(1+r^4)
v23(r) = -8/(1+r^5)
masses = [1.0,1.0,2.0]
pp = make_phys_params3B3D(;mass_arr = masses, vint_arr=[[v23],[v23],[v12]])

# Define numerical parameters
np = make_num_params3B3D(;gem_params=(nmax=10,r1=0.2,rnmax=20.0,Nmax=10,R1=0.2,
    RNmax=20.0))

# Solve the 3-body, 3D quantum system.
@time energies = ISGL_solve(pp,np) # ~0.5s on an average laptop
```

In the following subsections more physically relevant examples are discussed.

## 5.2 Two-body system: Coulomb interaction & resonances via complex scaling

Here, we showcase the usage of the module GEM2B. As a first detailed example, let us consider a two-body system in 3D with Coulomb interaction. The complete script file example3D.jl containing this example can be found in the examples folder of the **FewBodyToolkit.jl** repository and executed via include("examples/example3D.jl"). Let's go through it step by step. In the beginning, we must load all necessary packages.

```
using Printf, Plots, Antique, FewBodyToolkit
```

Our system is described by a named tuple containing the physical parameters. We can create such a named tuple by calling the function make_phys_params2B. For this system, we assume that one particle has mass 1.0, and the other is infinitely heavy, which effectively leads to a reduced mass of unity. As input, only the reduced mass is required. Since a value of 1.0 is the default, in principle we do not need to specify it explicitly. Moreover, we define the Coulomb interaction as a function v_coulomb with strength $Z = 1.0$. The interactions are collected in the argument vint_arr of the call to create the named tuple of physical parameters. Three dimensions and zero angular momentum ($s$-wave) are the default options and also do not need to be specified.

```
mass_arr = [1.0, Inf] # array of masses of particles [m1,m2]
mur = 1 / (1/mass_arr[1] + 1/mass_arr[2])
Z = 1.0

v_coulomb(r) = -Z/r

phys_params = make_phys_params2B(;mur=1.0, vint_arr=[v_coulomb],lmax=0,dim=3)
```

For the numerical parameters we choose to work with $n_{max} = 10$ basis functions with ranges in geometric progression, defined via $r_1 = 0.1$, $r_{nmax} = 30.0$. Then, a named tuple containing these numerical parameters is defined via a call to the function `make_num_params2B`.

```
nmax=10 # number of Gaussian basis functions
r1=0.1;rnmax=30.0; # r1 and rnmax defining the widths of the basis functions
gem_params = (;nmax,r1,rnmax);

num_params = make_num_params2B(;gem_params)
```

These two named tuples are sufficient for calling the solver `GEM2B_solve` to obtain the eigenenergies.

```
energies = GEM2B.GEM2B_solve(phys_params,num_params)
```

The Coulomb potential has infinitely many bound states, whose energies can be found exactly. We can use the package Antique.jl [45] to provide these energies as reference. A comparison is summarized in Tab. 2.

```
simax = min(lastindex(energies),6); # max state index for comparison

CTB = Antique.CoulombTwoBody(m₁=mass_arr[1], m₂=mass_arr[2])
energies_exact = [Antique.E(CTB,n=i) for i=1:40]
```

| Index | Numerical | Exact | Difference |
|-------|-----------|-----------|-----------|
| 1 | -0.499876 | -0.500000 | -0.000124 |
| 2 | -0.124543 | -0.125000 | -0.000457 |
| 3 | -0.054437 | -0.055556 | -0.001118 |
| 4 | -0.028644 | -0.031250 | -0.002606 |
| 5 | 0.007342 | -0.020000 | -0.027342 |
| 6 | 0.603176 | -0.013889 | -0.617065 |

Table 2: Numerical solution of the two-body Coulomb problem using non-optimized numerical parameters.

The numerical solutions are good for the few lowest states, but only four bound states are found. This is because the numerical parameters are not optimized. We can do that via the function `GEM_Optim_2B`, which takes an additional argument `stateindex`, indicating the state for which the numerical parameters shall be optimized. Note that optimizing the results for a specific state might lead to worse results for other states. The optimized Gaussian ranges are $r_1 = 0.871259$, $r_{nmax} = 45.664907$, and the corresponding results are listed in Tab. 3.

```
stateindex = 6
params_opt = GEM2B.GEM_Optim_2B(phys_params, num_params, stateindex)
gem_params_opt = (;nmax, r1 = params_opt[1], rnmax = params_opt[2])
num_params_opt = make_num_params2B(;gem_params=gem_params_opt)
energies_opt = GEM2B.GEM2B_solve(phys_params,num_params_opt)
```

Optimizing the parameters for the sixth excited state finds more bound states, while loosing some accuracy for the lower states. We emphasize that we obtain good results for six states using only ten basis functions. Now, only more basis functions would help.

| Index | Non-optimized | Optimized | Exact | Difference |
|-------|---------------|-----------|-------|------------|
| 1 | -0.499876 | -0.489956 | -0.500000 | -0.010044 |
| 2 | -0.124543 | -0.123714 | -0.125000 | -0.001286 |
| 3 | -0.054437 | -0.055167 | -0.055556 | -0.000388 |
| 4 | -0.028644 | -0.031063 | -0.031250 | -0.000187 |
| 5 | 0.007342 | -0.019847 | -0.020000 | -0.000153 |
| 6 | 0.603176 | -0.013823 | -0.013889 | -0.000066 |

Table 3: Optimizing the numerical parameters can greatly improve the accuracy.

Highly accurate results can indeed be obtained by using a larger basis. For a two-body system this comes only at a moderate computational cost. Here, we use complex-ranged Gaussian basis functions via the optional keyword argument `cr_bool=1`. These complex-ranged Gaussians enrich the basis functions by oscillatory features, which is useful for calculating highly-excited states or resonances. Tab. 4 shows a comparison of the results based on the following numerical parameters, to those of Table 2 of Ref. [19], and the exact ones. Accurate results are obtained up to $n = 40$.

```
np = make_num_params2B(;gem_params=(;nmax=80,r1=0.015,rnmax=2000.0),omega_cr=1.5
    ,threshold=10^-11)
@time energies_accurate = GEM2B.GEM2B_solve(phys_params,np;cr_bool=1) # ~2s on
    an average laptop
```

| Index | Complex-Ranged | Exact | Difference |
|-------|----------------|-------|------------|
| 1 | -0.500000 | -0.500000 | -0.000000 |
| 2 | -0.125000 | -0.125000 | -0.000000 |
| 3 | -0.055556 | -0.055556 | -0.000000 |
| 4 | -0.031250 | -0.031250 | -0.000000 |
| 5 | -0.020000 | -0.020000 | -0.000000 |
| 10 | -0.005000 | -0.005000 | -0.000000 |
| 14 | -0.002551 | -0.002551 | -0.000000 |
| 18 | -0.001543 | -0.001543 | -0.000000 |
| 22 | -0.001033 | -0.001033 | -0.000000 |
| 26 | -0.000740 | -0.000740 | 0.000000 |
| 30 | -0.000556 | -0.000556 | 0.000000 |
| 32 | -0.000488 | -0.000488 | -0.000000 |
| 34 | -0.000432 | -0.000433 | -0.000000 |
| 36 | -0.000386 | -0.000386 | 0.000000 |
| 38 | -0.000347 | -0.000346 | 0.000001 |
| 40 | -0.000311 | -0.000313 | -0.000001 |

Table 4: Accurate results for highly-excited states can be found with more basis function and employing complex-ranged basis function.

We can also calculate the wave functions and compare to the exact results which are again provided by the package Antique.jl. For that we use the additional argument `wf_bool=1` to output the eigenvectors which contain the amplitudes of the Gaussian basis functions. The function `wavefun_arr` then provides an array containing the values of the wave function in position space, evaluated at the grid points defined by `r_arr`.

```
energiesw,wfs = GEM2B.GEM2B_solve(phys_params,num_params_opt;wf_bool=1,cr_bool=0
    );

dr = 0.1
r_arr = 0.0:dr:50.0
redind = vcat(1:2:30,31:5:50,51:10:lastindex(r_arr))

wfA(r,n) = Antique.R(CTB, r; n, l=0) # Exact wave function for the n-th state

density = zeros(length(r_arr),4)
density_exact = zeros(length(r_arr),4)
for si = 1:4
    wf = wfs[:,si]
    psi_arr = GEM2B.wavefun_arr(r_arr,phys_params,num_params_opt,wf;cr_bool=0)

    density[:,si] .= abs2.(psi_arr).*r_arr.^2
    density_exact[:,si] = abs2.(wfA.(r_arr,si)).*r_arr.^2
end
```

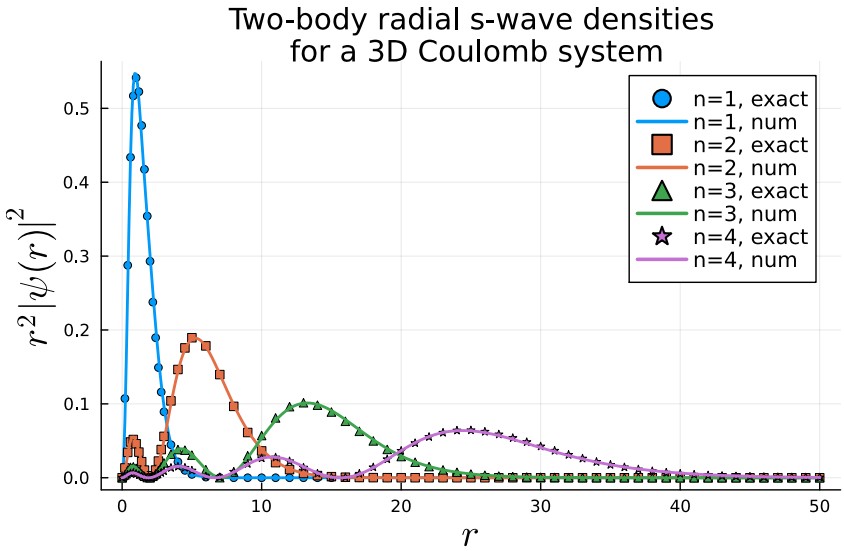

Figure 2: Densities of *s*-wave eigenstates of the Coulomb potential. Numerical solutions obtained via **FewBodyToolkit.jl** are shown as solid lines, exact values provided by Antique.jl as markers.

The resulting radial densities $r^2|\psi(r)|^2$ are displayed in Fig. 2 together with the analytical solutions indicated by various markers. We find very good agreement over the entire range. We can also check that the wave functions are properly normalized by integrating the density. A simple Riemann sum is sufficient here and yields $0.999999, 0.999999, 0.999997, 0.998430$.

```
norms = density[:,1:4]'*fill(dr,lastindex(r_arr))
```

To finalize the two-body discussion, we showcase the possibility to compute resonances. A script file `CSM2B.jl` can be found in the examples folder of the **FewBodyToolkit.jl** repository. For that we consider the potential

$$v(\lambda, r) = \frac{\lambda}{r} \left[ 678.1 \, e^{-2.55r} - 166.0 \, e^{-0.68r} \right] \tag{13}$$

discussed in Ref. [46] which, depending on the value of $\lambda$, has a single bound state or a single resonance. This potential is relevant for nuclear physics, since it allows for a qualitative

description of the experimental $p$-wave phase shifts of n-$^3$H scattering [46]. To be precise, $\lambda = 1$ reproduces the $J^\pi = 2^-$ case[1], whereas $\lambda = 0.75$ corresponds to $J^\pi = 0^-$.

```
function csm2b(lambda,nmax)
    function v(r)
        r > 50.0 && return 0.0
        return lambda*(678.1*exp(-2.55*r) - 166.0*exp(-0.68*r))/r
    end

    mur = 1/(2*27.647)
    pp = make_phys_params2B(;mur,vint_arr=[v],dim=3,lmin=1,lmax=1)
    np = make_num_params2B(;gem_params=(nmax=nmax, r1=0.3, rnmax=30.8),theta_csm
        =40.0)

    e2 = GEM2B_solve(pp,np,csm_bool=1)

    return e2
end
```

We define a function that computes the eigenspectrum for a given value of $\lambda$. Note the extra arguments `csm_bool` and `theta_csm` to enable complex scaling and to set its rotation angle. Then, the results for different values of $\lambda$ are computed in a loop. The resonances are filtered from the spectrum via a helper function `is_in_triangle` (see script file for details). The output is summarized in Tab. 5. With the use of the complex scaling option we can accurately reproduce both resonance position (real part) and width (imaginary part).

| $\lambda$ | Real | Reference | Difference | | $\lambda$ | Imag | Reference | Difference |
|---|---|---|---|---|---|---|---|---|
| 1.75 | -1.7914 | -1.7914 | -0.0000 | | 1.75 | -0.0000 | 0.0000 | 0.0000 |
| 1.50 | 0.0932 | 0.0933 | 0.0001 | | 1.50 | -0.0151 | -0.0152 | -0.0001 |
| 1.25 | 0.9713 | 0.9712 | -0.0001 | | 1.25 | -0.7446 | -0.7446 | -0.0000 |
| 1.00 | 1.2609 | 1.2670 | 0.0061 | | 1.00 | -1.9923 | -2.0020 | -0.0097 |

Table 5: Comparison of the real parts (left) and imaginary parts (right) of the resonance energy against the values of Ref. [46].

## 5.3 Mass-imbalanced 2+1 system in 1D

Next, we demonstrate how to use the module `GEM3B1D`. As a second example, we consider a three-body system in 1D, consisting of two identical particles, and a different one. The universality of this system was analyzed in Ref. [32] for an interspecies pair-interaction supporting a weakly bound ground state. Here, we showcase the calculation for the cases of a Gaussian and a contact interaction. The two identical particles do not interact. A complete script `GEM3B1D_2+1.jl` can be found in the examples folder of the **FewBodyToolkit.jl** repository.

First, we define the 2+1 system via the mass ratio. Here we choose the value 22.2 since it results in the most bound states. The Gaussian interaction is constructed via the exported type `GaussianPotential` taking the two arguments `v0`, and `mu_g` denoting the depth and width, respectively. To ensure a weakly bound ground state energy, we set the target energy to $10^{-3}$, and first solve the inverse *two-body* problem via the function `v0GEMOptim` of the module `GEM2B` to find the corresponding depth. The dimensionality is set via `dim=1` in the physical parameters.

```
using Printf, FewBodyToolkit
```

---

[1]In nuclear physics it is common to represent the total angular momentum $J$ and the parity $\pi$ in this notation.

```
massratio = 22.2 # other values are 2.2 and 12.4
mass_arr = [1.0, massratio, massratio]
mur = 1/(1/mass_arr[1]+1/mass_arr[2]) # reduced mass

v0 = -1.0; mu_g = 1.0;
vg = GaussianPotential(v0,mu_g)

phys_params2B = make_phys_params2B(;mur,vint_arr=[vg],dim=1)
num_params2B = make_num_params2B(;gem_params=(;nmax=6, r1=1.0, rnmax=20.0))

stateindex = 1; target_e2 = -1e-3;
pps,nps,vscale = GEM2B.v0GEMOptim(phys_params2B,num_params2B,stateindex,
    target_e2)
```

The value `vscale` is the required scaling of the original value of $v_0$ such that the desired two-body energy is found. With this, we can define the rescaled Gaussian potential which will be used for the three-body calculation. The two-body system has indeed the desired binding energy, `e2s` $= -0.001000000011 \approx 0.001$.

```
vgscaled = GaussianPotential(v0*vscale,mu_g)
pps = make_phys_params2B(;mur,vint_arr=[vgscaled],dim=1)

e2s = GEM2B.GEM2B_solve(pps,nps) # consistency check
```

For the contact interaction we don't need to find the potential strength, but parameters should be optimized

```
vc = ContactPotential1D(-sqrt(-2*target_e2),0.0)
ppc = make_phys_params2B(;mur,vint_arr=[vc],dim=1)
npc = make_num_params2B(;gem_params=(;nmax=16, r1=1.0, rnmax=120.0))

r1cs,rnmaxcs,e2copt=GEM_Optim_2B(ppc,npc,stateindex)
```

Having found the potential parameters, we can now set up the three-body problem with the scaled potential. For bosons we can make use of their symmetry with the argument `svals=["x","b","b"]`, where `x` is the different particle and `b` denotes two identical bosons. The code then automatically employs the appropriate symmetrization and reduces the number of Faddeev components to the minimal one. We use the optimized values for the numerical parameters `nmax`, `r1`, `rnmax`, found by the two-body inverse problem. The parameters for the other Jacobi coordinate are set manually.

```
vint_arr=[[],[vgscaled],[vgscaled]] #[[v23],[v31],[v12]]
phys_params3B = make_phys_params3B1D(;mass_arr=mass_arr,svals=["x","b","b"],
    vint_arr)

nmax = nps.gem_params.nmax;
r1 = nps.gem_params.r1; rnmax = nps.gem_params.rnmax;
num_params3B = make_num_params3B1D(;gem_params=(;nmax, r1, rnmax, Nmax=16, R1=1
    .5, RNmax=250.0))
```

Having defined the input parameters, we can go on and find the three-body energies by calling the function `GEM3B1D_solve`. Replacing `vgscaled` by `vc`, we can perform the calculation for the contact interaction in an analogous way (see script file for more details). Since Ref. [32] provides values of the ratio of three-body to two-body energies, we compute the ratios `epsilon`. A comparison of the results to the ones from the article can be seen in

Tab. 6.

```
e3 = GEM3B1D.GEM3B1D_solve(phys_params3B,num_params3B);

epsilon = e3 /abs(e2s[1]) # energy ratios
```

| | Bosons | | | Fermions | | |
|---|---|---|---|---|---|---|
| Index | Gaussian | Contact | Reference | Gaussian | Contact | Reference |
| 1 | -2.74274 | -2.75157 | -2.7515 | -1.69497 | -1.69048 | -1.6904 |
| 2 | -1.36058 | -1.36044 | -1.3604 | -1.14929 | -1.14795 | -1.1479 |
| 3 | -1.05240 | -1.05255 | -1.0525 | -1.00423 | -1.00403 | -1.0040 |

Table 6: Comparison of results from **FewBodyToolkit.jl** for Gaussian and contact interactions, with reference values of Ref. [32], for a mass ratio of 22.2 in the case the two identical particles are bosons (left), or fermions (right).

When the two identical particles are fermions, we can use the same potential. To account for their different statistics and parity, we use svals=["x","f","f"] and parity=−1. To allow for basis functions that obey these requirements (a node at vanishing distance), we need to set lmax, Lmax=1. Again, we also compute the results for the contact interaction. A comparison with the article's result is summarized in Tab. 6.

```
phys_params3B_F = make_phys_params3B1D(;mass_arr=mass_arr,svals=["x","f","f"],
    vint_arr,parity=-1)
num_params3B_F = make_num_params3B1D(;gem_params=(;nmax, r1, rnmax, Nmax=16, R1=
    1.5, RNmax=250.0), lmin=0, Lmin=0, lmax=1, Lmax=1)
e3_F = GEM3B1D.GEM3B1D_solve(phys_params3B_F,num_params3B_F)

epsilon_F = e3_F /abs(e2s[1])
```

Overall, we can reproduce the article's results very well for both bosonic and fermionic systems. Note that a finite discrepancy between the results using a Gaussian and a contact interaction is expected. Only in the limit of vanishing two-body binding energy, the two potentials should yield the same three-body results.

## 5.4 Positronium negative ion

Finally, we provide an example for the module ISGL. For that, let us discuss the positronium negative ion, Ps⁻, a system of three charged particles (electron, electron, positron) in 3D, interacting pairwise via Coulomb interactions. This system was analyzed to very high precision in Ref. [47]. In that article, not only the binding energies, but also the mean values of several observables (mean radii) are provided.

This three-body system can be solved with the module ISGL. The corresponding solver function ISGL_solve provides on-the-fly calculation of mean values of central observables via the optional argument observ_params. First, we define the interaction and masses of the particles. Since we work in atomic units, the electrons' and positron's masses are set to unity.

```
using Printf, FewBodyToolkit

# Define pair-interactions:
vee(r) = +1/r #electron-electron: V12; repulsive
```

```
vep(r) = -1/r #positron-electron: V31, V23; attractive

# Physical parameters
mass_arr=[1.0,1.0,1.0]
svals=["b","b","z"]
phys_params = make_phys_params3B3D(;mass_arr, svals, vint_arr=[[vep],[vep],[vee]
    ]);

# numerical parameters:
gp = (;nmax=10,Nmax=10,r1=0.1,rnmax=25.0,R1=0.1,RNmax=25.0)
num_params = make_num_params3B3D(;gem_params=gp);
```

In order to calculate the mean values of a central, i.e. only $r$-dependent, observable, we must provide an optional keyword argument `observ_params` (together with `wf_bool=1`), containing the state-indices for which the observables should be calculated (e.g. 1 for the ground state), and `centobs_arr`, a vector of vectors of observable-functions for each of the three Jacobi sets (similar to how the interactions are defined). Moreover, via `R2_arr`, we can decide whether $\langle R^2 \rangle$, the mean square distance of one particle relative to the center-of-mass of the other two, is computed for each of the Jacobi sets (1 indicates yes, 0 no).

The results for the energies are stored in the `energies` array, the eigenvectors in `wfs`, and the central observables in `co_out`. The mean squared radii for the R-coordinate are stored in `R2_out`.

```
rad(r) = r # radius
invrad(r) = 1/r # inverse radius
rad2(r) = r^2; # squared radius

stateindices = [1] # for which states to calculate observables
observ_params = (stateindices,centobs_arr = [[rad,invrad,rad2],[rad,invrad,rad2]
    ,[rad,invrad,rad2]],R2_arr = [1,1,1] ) # R2_arr=[0,0,0] means no R^2
    calculation

energies,wfs,co_out,R2_out = ISGL.ISGL_solve(phys_params,num_params;
    observ_params,wf_bool=1);
```

Via the helper function `comparison` we can then compare the results to those of the literature. The results are summarized in Tab 7.

| Observable | Numeric | Reference | Rel. difference (%) |
|---|---|---|---|
| $E$ | -0.261815 | -0.262005 | 0.073 |
| $\langle r_{pe} \rangle$ | 5.499094 | 5.489630 | -0.172 |
| $\langle r_{ee} \rangle$ | 8.542070 | 8.548580 | 0.076 |
| $\langle 1/r_{pe} \rangle$ | 0.339703 | 0.339820 | -0.034 |
| $\langle 1/r_{ee} \rangle$ | 0.155783 | 0.155630 | 0.099 |
| $\langle r_{pe}^2 \rangle$ | 48.633676 | 48.418900 | 0.445 |
| $\langle r_{ee}^2 \rangle$ | 93.050415 | 93.178600 | -0.148 |
| $\langle R_{e,ep}^2 \rangle$ | 58.6836 | - | - |
| $\langle R_{p,ee}^2 \rangle$ | 25.3711 | - | - |

Table 7: Comparison of numerical results with literature values of Ref. [47] for binding energy and various observables.

Since this system has only a single bound state, we can reproduce both energies and geometric properties with good accuracy. For the observable $\langle R^2 \rangle$, the reference does not provide any values, so we just print the numerical results.

# 6 Benchmarks

In this section, we provide benchmark calculations. The benchmarks were obtained on an Intel Xeon CPU E5-2697 v2 running Ubuntu 24.04 and Julia 1.10.4 on a single thread.

## 6.1 Performance scaling: Time and memory

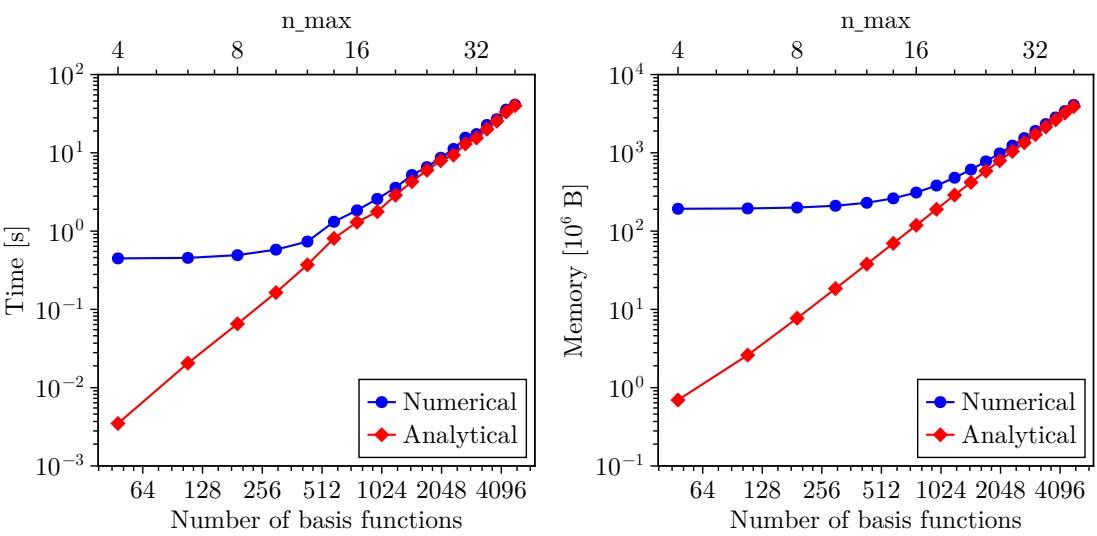

Figure 3: Scaling of runtime (left) and memory usage (right) with basis size $3n_{\max}^2$ in a double-logarithmic scale. We compare the total costs when evaluating the interaction matrix elements (i) numerically (blue), and (ii) using analytical expressions (red).

The performance of the package is benchmarked in terms of runtime and memory usage as a function of the basis size. As an example, we use the ISGL module and a Gaussian interaction (for more details see the examples/Benchmark subfolder in the **FewBodyToolkit.jl** repository).

```
vg(r) = -10.0*exp(-r^2)
pp = make_phys_params3B3D(;mass_arr=[1.0,2.0,3.0],svals=["x","y","z"],vint_arr=[
    [vg],[vg],[vg]])

vga = GaussianPotential(-10.0,1.0)
ppa = make_phys_params3B3D(;mass_arr=[1.0,2.0,3.0],svals=["x","y","z"],vint_arr=
    [[vga],[vga],[vga]])

nn=10
gp = (;nmax=nn,Nmax=nn,r1=0.1,rnmax=100.0,R1=0.1,RNmax=100.0)
np = make_num_params3B3D(;gem_params=gp,kmax_interpol=2000,lmin=0,lmax=0,Lmin=0,
    Lmax=0,threshold=10^-10)
```

As shown in Fig. 3, both time and memory scale polynomially (linear in the double-logarithmic scale) with the number of basis functions. The scaling remains well controlled up to several thousand basis functions, and the growth is sufficiently mild that realistic three-body problems can be treated with modest computational costs. Explicit values are listed in Tab. 8. Note that these values differ depending on the module and physical system at hand. Needless to say, they also depend on the specific hardware in use.

| $n_{\max}$ | Runtime [s] | Memory [$10^6$B] | Eigenvalue |
|---|---|---|---|
| 6 | 0.45 | 195 | $-11.620$ |
| 10 | 0.58 | 211 | $-14.349$ |
| 20 | 3.58 | 481 | $-14.435$ |
| 30 | 15.6 | 1540 | $-14.435$ |

Table 8: Benchmark values for runtime, memory usage, and the lowest eigenvalue. The values are obtained in the case of a numerical treatment of the interaction matrix elements.

## 6.2 Numerical vs analytical treatment of interaction

In Fig. 3 we also compare how total runtime and memory usage is affected by the calculation of the interaction matrix elements via (i) direct numerical integration with subsequent interpolation (see Sec. 4 for details), and (ii) using analytical expressions. For small basis sizes, the analytical evaluation reduces runtime and memory usage by up to two orders of magnitude, while for larger basis sizes the advantage diminishes. This is because at large `n_max` the interpolation scheme limits the costs for the numerical evaluation. Then, most of the computational effort is spent in the diagonalization step itself and the two approaches perform comparably for the largest systems considered.

# 7 Conclusion and Outlook

We have introduced **FewBodyToolkit.jl**, a Julia package for solving general quantum few-body problems. The package supports both two- and three-body systems in different spatial dimensions, allows for general forms of pairwise interactions, and can address bound as well as resonant states. It is easily installed via Julia's package manager and comes with documentation and examples. While two-body calculations already offer useful applications, providing a direct treatment of the richer and more demanding three-body problem is a core motivation for the package and its main research relevance. To our knowledge, no comparable open-source few-body solver with a similar feature set and documentation exists in Julia or elsewhere.

The current implementation across the provided modules is based on the well-established Gaussian expansion method. However, this approach can face challenges, for example in the presence of broad and strongly repulsive interactions. Possible extensions for this package might therefore implement other methods, such as momentum-space Faddeev integral equations, connections to the hyperspherical approach, or neural-network-based machine-learning techniques. More near-term extensions include the treatment of additional interaction types such as spin-dependent, tensor, three-body, and non-local forces. Future work may also include the computation of other observable quantities, such as scattering properties and phase shifts, or external potentials [48, 49]. While an extension to larger particle numbers beyond three-body calculations sounds natural, it likely requires an implementation that scales more generally with the number of particles, rather than separate implementations for systems of four, five, and more constituents.

We hope that **FewBodyToolkit.jl** can develop into a standard framework for quantum few-body calculations with Julia, providing a flexible and extensible basis for applications and method-development.

## Acknowledgements

We thank E. Hiyama, D. Yoshida, S. Ohno, N. Yamanaka, and S. Yoshida for fruitful discussions on the Gaussian expansion method and Julia package organization.

**Funding information**   L. H. is supported by the RIKEN special postdoctoral researcher program (SPDR).

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
