# Peer review of "FewBodyToolkit.jl: a Julia package for solving quantum few-body problems"

_SciPost Physics Codebases_

## Round 1 · Referee Report · Anonymous (Referee 1) · 2025-11-14

Disclosure of Generative AI use

The referee discloses that the following generative AI tools have been used in the preparation of this report:

I used Copilot to identify typos and improve the language in the report I had already written.

Strengths

General open-source software for two- and three-body problems

Weaknesses

The manuscript often reads like an internal technical note

Report

The manuscript introduces FewBodyToolkit.jl, a Julia-based software package for solving quantum two- and three-body problems. The toolkit implements the well-known Gaussian expansion method and includes a description of the code along with a discussion of several illustrative examples. Additional documentation and further examples are available in the repository.

A general open-source code for three-body problems is certainly valuable, and I believe this submission has a strong potential to be a useful addition to SciPost Phys Codebases. However, there are several issues that need to be addressed before the manuscript can be accepted for publication. My main concern is that, in its current form, the manuscript often reads like an internal technical note, which limits the potential usefulness of the software for a broader audience, which is apparently one of the main goals of the present toolkit (e.g., the manuscript states “we see further applications for teaching”). In "Requested changes" below, I provide a detailed list of comments and suggestions.

Requested changes

Major:

  1. The manuscript states that FewBodyToolkit.jl can calculate resonances, which is a potentially very useful feature. However, the manuscript does not explain how these calculations are actually performed. The manuscript should include a clear description of the complex scaling method used, as well as the relevant parameters that appear in the code. This is important because, when I attempted to use the code to find resonances in the well-studied “Hazi–Taylor” model (https://doi.org/10.1103/PhysRevA.1.1109), I found the results from FewBodyToolkit.jl to be highly unstable, with a strong dependence on the chosen parameters.

  2. The manuscript introduces several parameters (e.g., nmax, r1, rmax) without explaining the best strategies for choosing them. The manuscript should clearly define each parameter, describe its significance, and provide guidance on how to select appropriate values. This addition will also clarify some of the presented results, e.g., the relation between nmax and the number of functions in Figure 3.

  3. The manuscript states: “To our knowledge, no general few-body solver with both accessibility and comprehensive documentation is currently available.” However, even a quick ChatGPT search suggests that several packages exist, such as Rimu.jl, FewBodyPhysics.jl, and FermiFCI.jl. While their documentation varies in accessibility, these codes are available and are capable of addressing certain few-body problems. Therefore, a more thorough review of the existing open-source software, accompanied by a brief comparison, is necessary.

  4. The repository includes several examples that are not mentioned in the manuscript. It would be helpful to add a table listing these examples along with a brief description of the corresponding physical systems.

  5. The manuscript assigns the discrepancy in Table 6 between “Gaussian” and “Contact” data to the breakdown of universality, see “Only in the limit of vanishing two-body binding energy, the two potentials should yield the same three-body results.” However, to meaningfully discuss the difference between “Gaussian” and “Contact,” it is essential to ensure that the results are converged to within sub-percent accuracy. Is this the case for the data presented in Table 6? More generally, it would be helpful if the manuscript addressed the accuracy of the reported results and explained how readers can estimate error bars using FewBodyToolkit.jl.

Minor:

A. It would greatly help readers appreciate the capabilities of FewBodyToolkit.jl if the manuscript outlined a few state-of-the-art problems that can be solved using the toolkit, as well as those that cannot. The manuscript only states “However, this approach can face challenges, for example in the presence of broad and strongly repulsive interactions.”. This statement is however too vague and some examples would be appreciated. This could be achieved by discussing a few recently addressed few-body problems in the context of the software’s functionality and limitations.

B. Overall, the manuscript frequently lacks references in particular in Secs. 2 and 7. For example, the sentence “Possible extensions for this package might therefore implement other methods, such as momentum-space Faddeev integral equations, connections to the hyperspherical approach, or neural-network-based machine-learning techniques” does not contain too much information without references. Adding a few references would help guide newcomers to the field of few-body research.

C. The manuscript states “complete script GEM3B1D_2+1.jl can be found in the…” I could not find it.

D. In the second sentence of Sec. 5.1, what is meant by “As a second example”?

E. The captions of Tables 3,4,6,7 are not very informative. It would be helpful if the captions contained information about the considered system and what is presented.

Recommendation

Ask for major revision

---

## Editorial Decision

in_refereeing